# Postpartum Interventions to Increase Maternal Vaccination Uptake: Is It Worth It?

**DOI:** 10.3390/vaccines12101130

**Published:** 2024-10-01

**Authors:** Eleni Konstantinou, Sofia Benou, Eleftheria Hatzidaki, Aggeliki Vervenioti, Gabriel Dimitriou, Vassiliki Papaevangelou, Christine E. Jones, Despoina Gkentzi

**Affiliations:** 1Department of Paediatrics, Patras Medical School, 26504 Rio Achaia, Greece; elenikonstantinou19@gmail.com (E.K.); sofmpenou@gmail.com (S.B.); aggelikivervenioti@gmail.com (A.V.); gdim@upatras.gr (G.D.); 2Department of Neonatology and Neonatal Intensive Care Unit, School of Medicine, University of Crete, University Hospital of Heraklion, 71003 Heraklion, Greece; el.hatzidaki@uoc.com; 3Pediatric Infectious Diseases Department, Third Department of Pediatrics, National and Kapodistrian University of Athens, 15772 Athens, Greece; vpapaev@gmail.com; 4Faculty of Medicine, Institute for Life Sciences, University of Southampton, Southampton SO16 6YD, UK; c.e.jones@soton.ac.uk; 5NIHR Southampton Clinical Research Facility and NIHR Southampton Biomedical Research Centre, University Hospital Southampton NHS Foundation Trust, Southampton SO16 6YD, UK

**Keywords:** vaccination, postpartum, maternal

## Abstract

Background/Objectives: Vaccination of pregnant and postpartum women for pertussis, influenza and COVID-19 not only protects themselves but also offspring. Despite the benefits of this approach, vaccination uptake remains suboptimal in pregnancy. Where the opportunity to be vaccinated in pregnancy is missed, the offer of vaccination in the post-partum period may be an alternative strategy. The aim of this systematic review is to assess the impact of interventions to increase vaccination uptake in the postpartum period on vaccination uptake. Methods: A literature search was performed in MEDLINE, including interventional studies promoting vaccination uptake in postpartum women published between 2009 and 2024. The search was conducted according to PRISMA guidelines and registered with PROSPERO. Results: We finally included 16 studies in the review, and the primary outcome was vaccination uptake in the postpartum period. The most significant factors for increasing uptake were recommendation from healthcare providers, type of interventions used, and delivery of vaccines in the maternity wards or the community. Conclusions: In conclusion, maternal vaccination rates in the postpartum period may increase with targeted education by healthcare professionals and positive reinforcement. The interventions described in these studies could be applied in the healthcare systems worldwide.

## 1. Introduction

Pertussis, influenza, and COVID-19 are three highly contagious infectious diseases transmitted by close contact through respiratory droplets. Pertussis can cause severe disease and even death in neonates and young infants [1,2,3,4]. Influenza and COVID-19 may also cause serious illness in pregnant women as well as neonates and young infants [5,6,7,8,9]. Vaccination in pregnancy against each of these infections is widely recommended around the world, and where this opportunity is missed, postpartum vaccination may be employed to directly protect women and indirectly protect their offspring.

For pertussis, the current primary pediatric vaccination series start at the minimum age of six weeks [2,10], which leaves a window of vulnerability for neonates and young infants, who may contract the disease from adult household members [1]. Vaccinating pregnant women with the combined tetanus toxoid, reduced diphtheria toxoid, and acellular pertussis vaccine (Tdap) between 27 and 36 weeks of gestation is highly effective at preventing hospitalization from pertussis in newborns by facilitating transplacental antibody transfer to the fetus [2,3,11]. Since 2006, the Advisory Committee of Immunization Practices (ACIP) has recommended Tdap in the immediate postpartum period for women not vaccinated during pregnancy and for all household contacts of the newborn to indirectly protect the infant by reducing the infection risk amongst adults caring for the newborn and via breast milk transfer of antibodies. In 2012, the ACIP updated their advice, offering Tdap in every pregnancy, regardless of previous vaccination, preferably at 27–36 weeks of gestation [1,12]. If Tdap vaccination has not been given during pregnancy, postpartum women are encouraged to be vaccinated [12,13].

Similarly to pertussis, influenza and Severe Acute Respiratory Syndrome Coronavirus 2 (SARS-CoV-2) are associated with significant morbidity and mortality in specific high-risk populations, such as infants and pregnant women [5,6,7,8,9]. Maternal influenza and COVID-19 vaccination benefit both the mother and their offspring and reduce the risk of severe infection and hospitalization in neonates and young infants [5,6,14,15,16,17,18]. Influenza vaccines are not indicated in infants younger than six months [6,14,19]. The World Health Organization (WHO), the Advisory Committee on Immunization Practices (ACIP), and the American College of Obstetricians and Gynecologists (ACOG) recommend seasonal immunization with the inactivated influenza vaccine (IIV) for all pregnant women, regardless of the stage of pregnancy, as well as for postpartum women and caregivers of neonates and young infants [5,20]. As for COVID-19, the Centers for Disease Control and Prevention and other professional organizations such as the ACOG highly recommend that pregnant and lactating women receive immunization against COVID-19 [21].

Despite the importance of vaccination for pregnant and postpartum women and the current guidelines, vaccination coverage among pregnant women has not been ideal. Data from the Centers for Disease Control and Prevention (CDC) from 2023 showed a 55.4% vaccination rate for Tdap during pregnancy and a 47.2% vaccination rate for influenza before or during pregnancy, and 27.3% reported receipt of a COVID-19 bivalent booster dose before or during the current pregnancy [22]. Other studies from 2022 show similar COVID-19 vaccination rates in pregnancy, ranging from 27 to 31% [23,24]. Moreover, high levels of vaccine hesitancy have been reported across high-income countries, and several studies have examined the reasons underlining this trend for pertussis [25,26], influenza [25,26] and COVID-19 [26,27,28,29]. The most significant factors associated with maternal vaccine hesitancy were found to be concerns about vaccine safety and efficacy, lack of knowledge, fear of adverse effects, low perception of disease severity and sociodemographic characteristics [26,27].

In recent decades, efforts have been made and strategies have been implemented to increase maternal vaccination. This systematic review aims to summarize the literature specifically on the interventions during the postpartum period to improve maternal vaccination and indirectly protect newborns and young infants.

## 2. Materials and Methods

### 2.1. Search Strategy

This systematic review included interventional trials to promote vaccination uptake in postpartum women. It was conducted according to the PRISMA guidelines [30]. The literature search was performed in MEDLINE, and the final search was performed on 31 January 2024. The literature search was conducted according to the PICO framework. The different components used for our search were “postpartum women” for population, “intervention” for exposure, and “vaccination” for outcome. We did not include comparison or control in the search keywords.

The following keywords and combinations of these were used for the search: “postpartum,” “intervention”, “maternal”, and “vaccination”. Snowball searching was performed to search for further relevant articles in the reference list of included articles. The study was registered with PROSPERO (registration number: CRD42024493910).

### 2.2. Eligibility Criteria

Two unblinded reviewers (E.K. and S.B.) were assigned to screen the title and abstract of each study to see if they met the eligibility criteria. Any disagreements were discussed with a third reviewer (DG). The eligibility criteria included interventional studies (randomized or not) that involved human participants published in English from January 2009 to January 2024 to promote postpartum vaccination.

Studies including data about pertussis/influenza/COVID-19 but irrelevant to vaccination were excluded from this review. Studies including data about vaccination uptake in the postpartum period but not including some types of intervention were excluded. Intervention trials not aiming to increase vaccination uptake in the postpartum period were also excluded.

The studies included aimed to promote vaccination with pertussis, influenza, COVID-19 or a combination of these vaccines. The populations in the studies were postpartum women and other household contacts of the newborns. The interventions were applied to this population in the maternity ward before discharge or in their first postpartum visits or in their first visits to the pediatric office soon after discharge from the maternity ward. Studies that included only pregnant women or other populations were excluded.

The intervention applied in the studies included information and education of the postpartum women, offer of the vaccine to postpartum women and other household contacts of the newborns, education of healthcare providers caring for postpartum women and newborns and any other measures applied promoting postpartum vaccination.

The outcome of the studies included were differences observed in vaccination uptake rates between the intervention and control arm or between the pre- and post-intervention periods. Supplemental outcomes were the reasons for vaccine hesitancy.

The process of exclusion of the non-eligible studies is presented in Figure 1.

### 2.3. Selection Process

The full text of papers meeting the inclusion criteria and none of the exclusion criteria was reviewed by two reviewers for inclusion. The de-duplication was performed manually. Data were extracted using a standardized data capture form. The Grading of Recommendations Assessment, Development and Evaluation (GRADE) was performed (Appendix A).

### 2.4. Strengths and Limitations

The present review has strengths and limitations. The main strength is that this is the first systematic approach in the field that may be used as a starting point for future interventional studies. As for the limitations, although we performed a quality assessment of the studies included with GRADE, one cannot exclusively rule out risks of bias, such as selection bias, inadequate blinding, selective outcome reporting or publication bias. Moreover, our search was performed in MEDLINE; hence, we might have missed studies published in other databases. Finally, the COVID-19 vaccine was licensed for use in 2020; therefore, limited data are available in the field compared to the other two vaccines recommended for years during pregnancy and postnatal period.

## 3. Results

### 3.1. Study Selection

A total of 490 studies were identified; of these, 458 were excluded based on title screening and abstract (Figure 1). The full text of the remaining 32 articles was screened for eligibility. Eighteen articles were excluded because they contained other types of intervention (not promoting maternal vaccination), did not contain original data, or the population was irrelevant. After a manual search of reference lists, two additional articles were obtained. We eventually included 16 interventional studies. All included studies were published between 2009 and 2023. Nine studies were conducted in the USA, two in France, one in Australia, one in Canada, one in Greece, one in Jordan and one in Taiwan. Of the two that were randomized [31,32], one was a single-arm trial [33], and one was a multi-arm intervention trial [34]. Study populations included women in the early postpartum period, whereas some also referred to fathers and other caregivers of newborns.

Eleven studies included interventions for the Tdap or Tdap-IPV vaccine (Table 1), three referred to the seasonal influenza vaccine (Table 2), and two to the COVID-19 vaccine [35,36] (Table 2).

In Table 1 and Table 2, we present the variables included in each article such as the number of participants and their characteristics, whether a questionnaire was completed prior to the intervention process, and the type of intervention applied to each one of these studies. Most importantly, the effects of these interventions on the vaccination uptake in the postpartum period are also presented. A summary of the 16 studies is provided below. The studies are mainly discussed and categorized according to the primary outcomes and the type of interventions applied. Finally, some supplemental outcomes of the studies are discussed.

### 3.2. Summary of Studies According the Results and Type of Intervention

#### 3.2.1. Primary Outcome According to Study Design

In the included studies, an intervention was applied to promote and increase vaccination uptake. The primary outcome was mainly the difference in vaccination uptake between the pre- and post-intervention period or between the intervention and control arm. In the observational study by Clarke et al., the effect of the intervention of 17 pharmacy students on the vaccination rates of postpartum women with Tdap was assessed. Immunization rates following counseling increased from 43.7% to 62.3% [40]. Similarly, Bernstein et al. conducted a quality improvement intervention trial in which they assessed the effect of a five-step intervention on the vaccination rate of eligible postpartum women compared to the pre-intervention baseline data [41]. In the pre-intervention period, 91 of the 166 eligible women (55%) received the vaccine, compared to 462 out of the 632 eligible women (73%) in the post-intervention period. Hence, the intervention led to an increase of 33% in the percentage of postpartum women who received the Tdap vaccine [41].

In some studies, the primary outcome was the comparison of vaccine uptake between the control and the intervention arm. For instance, the study conducted in Jordan by Momani et al. assessed the effect of a tele-educational program on increasing COVID-19 vaccination uptake in lactating, pregnant and women planning for pregnancy. In the intervention group, 162 out of 205 women (79%) received the vaccine, compared to 4 out of 220 (2%) in the control group. However, the number of postpartum women is not documented in this publication and the results are presented for the whole cohort [35].

In other studies, the primary outcome was the vaccine acceptance rate. An example is the decision-making interventional study conducted by Cheng et al., who provided a thorough multilevel education about maternal Tdap vaccination and offered Tdap vaccine before hospital discharge. In this study, among the 1207 participating postpartum women, 639 (53%) were vaccinated and 568 (46%) refused vaccination [37]. In addition, Leboucher et al. evaluated the impact of an intervention in the vaccination uptake of Tdap-IPV in the postpartum period by mothers and fathers. The overall vaccination rate was 69% (655/956) for mothers and 63% (584/931) for fathers [38].

As for COVID-19 vaccination, an interventional study was published by Kouba et al. The study was conducted between May and September 2021 and the researchers offered the COVID-19 vaccine to 8281 unvaccinated postpartum women before discharge from the maternity ward. Only 412 of them received the vaccine (5%) [36].

#### 3.2.2. Type of Intervention—The Role of Different Healthcare Professionals Providing Information

Healthcare providers play a significant role in the education of mothers. In the majority of the studies, physicians, midwifes and nurses were responsible for providing the postnatal education [33,37,38,41,44].

In some other studies, it is not clear whether the information was given to the postpartum women by a nurse or a physician. For instance, in the Frere et al. study, a “research assistant” was responsible for educating postpartum women and recommending the Tdap vaccine [34]. On the other hand, in the trial conducted by Hebbali et al., there was a core research team consisting of a pediatrician with expertise in infectious diseases, a pediatric surgeon, a nurse, and a research associate, and information could be provided by any of the members of the survey administration team [45]. Finally, in the study by Clarke et al., 17 pharmacy students, with the assistance of nurses and other healthcare professionals, informed and educated the postpartum women and encouraged them to be immunized with Tdap [40].

#### 3.2.3. Type of Intervention—The Role of Different Methods of Educational Process

The methods used for educating postpartum women included written information [33,37,38,39,40,41,42,43,44,45], posters [33,37,44], an educational video [37], oral education [33,34,37,38,39,40,41,42,43,44,45,46] and tele-education [35] provided by healthcare professionals.

A novel approach was used in the controlled interventional trial by Hayles et al. where participants were assigned to receive a gain-framed, loss-framed, or control education using weekly sequential block allocation [39]. The gain–loss message framing intervention indicated what mothers may gain from the postpartum vaccination strategy (gain) or what they may lose by not engaging with vaccination (loss). The control group received information from the 2009 New South Wales Health pertussis factsheet. Among eligible mothers, 70% (754/1080) were vaccinated post-intervention. Rates were similar between ‘gain’ (69.1%), ‘loss’ (71.8%) or ‘control’ (68.8%) pamphlets [39].

#### 3.2.4. Type of Intervention—The Role of Healthcare Provider’s Education as a Method of Intervention

A few studies also looked at the effects of the provider’s education on vaccine uptake. An interesting approach was the one used in the study by Clarke et al., who conducted a study focusing on the education and training of pharmacy students according to the CDC recommendations on Tdap immunization [40]. The trained students started educating and informing postpartum women as well as recommending vaccination. If women did not consent to the vaccine, an experienced healthcare professional continued to address immunization issues [40]. During the pre-intervention period the overall immunization rate in this postpartum unit was 43.7% (1116/2667). During the intervention period, 2411 women gave birth in the same maternity ward, and 1503 of them were vaccinated with Tdap (62.3%) [40]. Additionally, in other studies, educating healthcare professionals (nurses and targeted physicians) was the first step of an intervention promoting maternal vaccination [33,37,41,44]. This education was offered via participation in grand rounds and small group sessions. The role of education was to remind basic information about vaccination and provide updates in the field [33,37,41,44]. The proper education of physicians, midwifes, nurses and other researchers conducting the interventions is essential for the proper transmission of the information to the mothers and fathers. All the above studies showed that the education of the healthcare professionals is a first fundamental step of multilevel interventions that result in a significant increase in the vaccination rates.

#### 3.2.5. Type of Intervention—The Role of Vaccine Offer

In addition to the education provided to mothers and other family members, the intervention in all of the studies was the actual vaccine offer to participants either after discharge, in the maternity wards, or both. In some studies, participants were given a prescription for the specific vaccine. An example is the study conducted in France by Leboucher et al. in which all mothers and fathers received oral and written information about pertussis and the benefits and recommendations of Tdap-IPV vaccination. Afterwards, each parent received a prescription for Tdap-IPV at discharge, and a telephone interview was conducted two months postpartum. The prescription of the vaccine resulted in a vaccination coverage of 67.9% (267/393) in 2008 and 68.9% (388/563) in 2009 for mothers. The vaccination coverage of fathers was 63.1% (245/388) in 2008 and 62.4% (339/543) in 2009. In this study, a pre-intervention vaccination rate was not captured, but the authors comment on a previously reported lower vaccination rate in France (11.8%) [38]. Another example is the one by Bucchiotty et al., who conducted a “before” and “after” comparative study [43]. The “before” population was postpartum women and their partners approached in 2011, and the “after” population was postpartum women and their partners approached in 2015. Oral and written information was provided to all participants, but only in the “after” population was a prescription of the Tdap-IPV provided at discharge to participants, and a telephone interview was conducted 8–10 weeks after discharge [42,43]. In the “before” period (2011), 11 out of 64 mothers (17%) and 18 out of 68 fathers (27%) were vaccinated. In the “after” period, 54 out of 130 mothers (42%) and 39 out 136 fathers (29%) received the Tdap-IPV vaccine. This showed that except from the oral and written educational information provided, the prescription of the vaccine resulted in much higher vaccination rates, but only for mothers (17 vs. 42%, *p* < 0.001). Moreover, Walter et al. assessed the acceptance of Tdap vaccination among parents bringing their newborn to a pediatric office during the first month of life. Firstly, parents were informed verbally and in writing about the Tdap vaccination, and afterwards, parents eligible to receive the vaccine were encouraged to receive it in the pediatric office. Of the 160 eligible recipients of the Tdap vaccine, 82 (51.2%) eventually received a dose. Around 40% of them did so at a subsequent office visit occurring during the baby’s first month of life. The study concluded that implementing Tdap vaccination in the pediatric office increased the vaccination rate and is an option in cases where hospital-based, postpartum Tdap vaccination was not a routine practice [42].

In most studies, the vaccination was offered on the maternity ward before discharge, with maternal vaccination rates ranging from 53% to 72%, depending on the study [33,37,39,40,45,46].

#### 3.2.6. Supplemental Outcomes—Reasons for Vaccine Hesitancy

In addition to the vaccination rates, the reasons for not accepting vaccination and the factors influencing vaccination decisions were assessed in many studies. In a study from Canada, women reported that the main reasons for not accepting the Tdap vaccine were the need for more decision time (26.9%) and the fear of adverse effects (19.4%) [34]. Similarly, in France, the most common reasons for not being vaccinated during the postpartum period were lack of time, previous Tdap-IPV vaccination, and forgetting about it [38]. In a large US study, Healy et al. reported that Tdap refused because of fear of local reactions, concurrent medical conditions not stated as medical contradictions to vaccination, religious objections, and uncertainty about having received the vaccine recently [33]. In addition, in a smaller US study, Walter et al. recognized concerns about possible adverse events, uncertainty about the necessity of vaccination, and not feeling well after a cesarean section as the main reasons for vaccine refusal [42]. Finally, among the most common reasons behind vaccine refusal worldwide was the healthcare professional’s recommendation [37,39,45].

## 4. Discussion

This systematic review aimed to summarize what measures and interventions to increase vaccination in the postpartum period have been implemented in studies published so far. In addition, reasons for not accepting vaccination despite the interventions have been presented including vaccine hesitancy, misinformation propagated in the community, and lack of knowledge about maternal vaccination. The studies published and described above aimed to overcome these barriers and promote maternal vaccination in the postpartum period.

For influenza and Tdap, the vaccination rates after the proper intervention, as presented in the studies, are promising, ranging from 38% to 75% in most studies [32,33,34,35,37,38,39,42,43,44,45,46]. These results were independent of the study design. As for COVID-19, the two interventional studies included presented dissimilar results. The first study published in 2022 resulted in a very low vaccination rate (5%) [36], and the second published in 2023 resulted in 79% receipt of the vaccine in the intervention group, compared to 2% for the control group. Low vaccination rates globally, especially among pregnant women for the COVID-19 vaccine, highlight the need for further research on the promotion of this vaccine in pregnancy and the postpartum period.

The structured attempts presented in these 16 studies indicate that vaccination rates in this distinct population may increase with proper, purposeful intervention. Another aspect is the type of intervention implemented in each of these studies and the impact on vaccination rates and attitude changes. Patient education campaigns were very effective at increasing maternal vaccination rates. An example is the study conducted by Cheng et al., where education including video, oral, and written information and posters was used, with the outcome being a 53% Tdap vaccination rate [37]. In addition, the on-site vaccination in the maternity ward had a significant impact and could be implemented as a strategy, as proved, for example, in the studies conducted by Maltezou et al. (73.8% influenza vaccination rate) [46] and Healy et al. (75% Tdap vaccination rate) [33]. Multi-level interventions were also found to be effective and include, amongst others, patient education, actual vaccine offer, and mobilization of many aspects of the healthcare system, such as ensuring vaccine stock at the hospitals and providing the vaccine free of charge to the mothers and other family members. This type of intervention was applied by Bernstein et al., resulting in a 33% increase in postpartum Tdap immunization rates in the postintervention period [41]. The education of all healthcare professionals involved in the perinatal care is also of great importance. It is the first step in ensuring an efficacious multilevel intervention. When adequately informed and educated about maternal vaccination, they can educate the mothers more efficiently. In two studies in which healthcare providers’ education preceded the education of the postpartum women and the other family members, vaccination rates were exceptional (72–75% vaccination rates) [33,44].

As presented above, the most successful intervention is the one combining proper education of the healthcare professionals caring for postpartum women and the immediate offer and receipt of the vaccines. These results should be taken into account by professional organizations and healthcare professionals and should be implemented in the battle for reducing vaccine hesitancy and successfully promoting vaccination with Tdap, influenza and COVID-19 in pregnancy and the postpartum period. Obstetricians and other antenatal healthcare providers should first receive proper education and then promote the benefits of vaccines to pregnant women, addressing any misperceptions during each clinic visit. If the opportunity to do so antenatally is missed, then pediatricians, midwives, health visitors, general practitioners, and other healthcare practitioners ought to do so during postpartum visits. The physician’s recommendation is an essential determinant of maternal vaccination. Besides the physicians, the whole healthcare team should promote and deliver maternal vaccination as well as vaccination during the postpartum period. By adopting these measures, an increase in vaccination uptake could be achieved to provide optimal protection for these vulnerable populations. Further work is needed to determine how to sustain these changes in the longer term. The lessons learned are of great importance in light of the introduction of new antenatal vaccines that will further reduce morbidity and mortality in early infancy.

## Figures and Tables

**Figure 1 vaccines-12-01130-f001:**
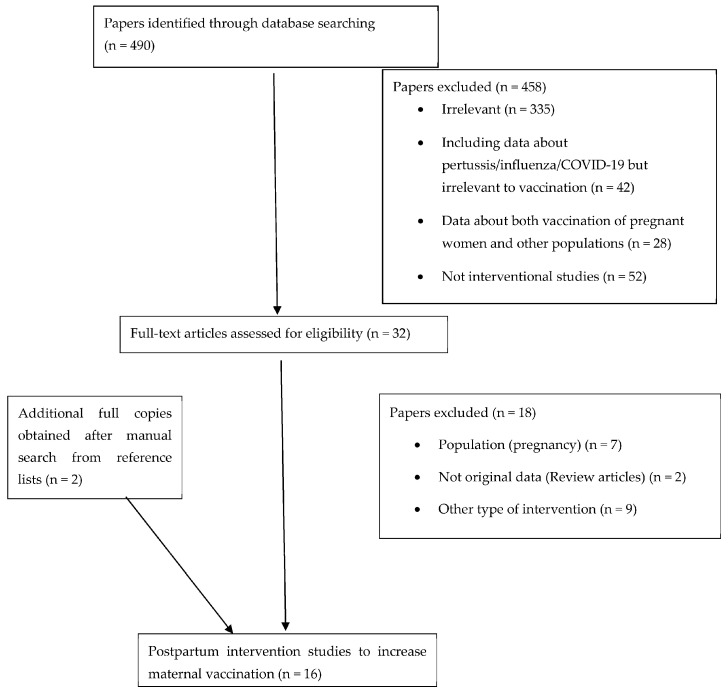
Study flowchart to identify and select eligible studies in the systematic review.

**Table 1 vaccines-12-01130-t001:** Studies presenting postpartum interventions to increase maternal vaccination and vaccinations of the other household members of the newborns with Tdap/Tdap-IPV.

Reference	Country /Year of Publication	Vaccine	Number of Participants and Characteristics	Study Design:Survey/Questionnaire	Intervention	Outcome
Cheng et al. [37]	Taiwan, 2010	Tdap ^1^	1241 women with uncomplicated delivery (2009)	Decision-making observational study. At first postpartum visit, 25 multiple-choice questions on: - Contagiousness of pertussis; - Effectiveness of Tdap; - Safety of the vaccine; - Perception of adequacy of the information received; - Factors influenced their decision to decline/accept the vaccine.	Information provided to all the participants during pregnancy about Tdap vaccination: - Posters; - Packet with educational material; - Educational video; - Information given orally by a trained nurse. Offer of the vaccine prior to hospital discharge.	53% accepted Tdap vaccination.
Leboucher et al. [38]	France, 2012	Tdap-IPV ^2^	659 postpartum women for Period A (January to March 2008) and 772 women for Period B (January to April 2009).	Prospective single-center observational study. No questionnaire used	- All mothers and also fathers (when present) given oral and written information about pertussis and Tdap-IPV vaccination; - Each parent received a prescription for Tdap-IPV vaccine before hospital discharge; - Two months postpartum, mothers were interviewed via telephone.	During Period A, 67.9% of mothers and 63.1% of fathers were vaccinated; During Period B, 68.9% of mothers and 62.4% of fathers were vaccinated.
Hayles et al. [39]	Australia 2014	Tdap	1404 postpartum women from a maternity hospital of Sydney from November 2010 to July 2012.	Controlled intervention trial. At 0–3 days postpartum, a baseline questionnaire concerning attitudes and beliefs about pertussis and Tdap vaccination was completed.	- Participants were assigned to receive a gain-framed, loss-frames or control. The gain–loss message framing intervention indicated either what mothers may gain from the cocooning strategy (gain) or what they may lose by not engaging in the cocooning strategy (loss); - The control group received information from the 2009 NSW ^3^ Health pertussis factsheet; - Offer of the Tdap vaccine to the participants.	70% of mothers were vaccinated post-intervention. Rates were similar between ‘gain’, ‘loss’ or ‘control’ groups. Overall pertussis immunization coverage increased from 23% to 77% among women screened.
Yeh et al. [32]	USA, 2013	Tdap	1252 postpartum women, 648 from the intervention hospital and 605 from the comparison hospital from October 2009 through July 2010.	Prospective controlled trial questionnaire on demographics and prior to receipt of Tdap	(A) Intervention hospital: (a) Opt-in order: providers had to check the order for women before hospital discharge. (b) Standing order: nurses delivered influenza and/or Tdap vaccines without additional order from the physician. (B) Control hospital: standard practice.	(A) Intervention hospital: (a) Opt-in order policy:18% increase in postpartum Tdap vaccination. (b) Standing order: further increase to 69%. (B) No postpartum Tdap vaccinations in the comparison hospital.
Healy et al. [33]	USA, 2009	Tdap	1570 postpartum (medically underserved, uninsured) women. January–April 2008	Single-arm interventional study. No questionnaire	- Education of the healthcare providers caring for postpartum women. - Education of postpartum women: information packet posters. - Postpartum Tdap vaccination recommended and offered to all women. - Vaccination of the women who consented before discharge.	- 72% of the participants received Tdap; - After excluding 396 women who were not offered the vaccine (history of recent receipt of Tdap, medical contraindications, no order written), Tdap uptake was 96.2%.
Frere et al. [34]	Canada, 2013	Tdap	345 postpartum women - 101 participants from September-October 2010 for Phase I - 244 participants from January-July 2011 for Phase II.	Multi-arm intervention trial- During Phase I, participants completed a questionnaire regarding: - pertussis knowledge; - attitudes; - immunization status. In phase II, no questionnaire was provided.	(A) Phase I: - Information about pertussis and cocooning strategy were provided; - Recommendation to be vaccinated in the community; - Contact by telephone six months later. (B) Phase II: - Information provided as in Phase I. In Phase II, Tdap vaccination offered in the maternity ward before discharge.	- Baseline knowledge was poor (6% of women protected) - In Phase I, 5.4% of mothers and 8.7% of fathers were immunized in the community; - In Phase II, 46.9% of mothers and 60.5% of fathers were immunized.
Clarke et al. [40]	USA, 2013	Tdap	A total of 1.263 postpartum women were consulted by the pharmacy students.	Observational study. No questionnaire	- Verbal and written information regarding pertussis immunization provided to mothers and caregivers of the newborns by 17 educated pharmacy students or by the nursing staff. - Offer of the vaccine to the participants - Vaccination before discharge to those who consented.	Following counseling, immunization rates, as a percentage of total births, significantly increased by 18.5%.
Bernstein et al. [41]	USA, 2017	Tdap	- Pre intervention baseline data (202 postpartum women, 166 eligible to receive Tdap). - Post intervention period: 844 women, of which 632 eligible for Tdap (August to December 2012)	Quality improvement intervention trial. No questionnaire	5-step intervention: - First step: nurse-driven education of all mothers regarding pertussis and Tdap vaccine (verbal and written) - Second step: offer of vaccination to each mother. - Third step: a standing order was created for Tdap vaccination during hospitalization. - Fourth step: keeping Tdap at floor stock. - Fifth step: document administration.	Increase by 33% in the postpartum mothers that received the Tdap vaccine before discharge in the postintervention period.
Walter et al. [42]	USA, 2009	Tdap	200 parents whose newborns received medical care during the first month of life (5 month intervention in 2007)	Observational study. No questionnaire	Parents were informed about the study and Tdap vaccination (verbal and written). Offer of vaccination to all eligible parents in the pediatric office.	Of the 160 eligible to receive Tdap vaccine, 82 (51.2%) received a dose.
Bucchiotty et al. [43]	France, 2021	Tdap-IPV	Before: 134 postpartum women (September 2011) After: 347 postpartum women (March-April 2015)	Before-and-after comparative study. During pregnancy, the participants each filled out a questionnaire to report their immunization status.	Oral and written information was provided in the “before” and in the “after” period. In the “after” period: before discharge all women who were unimmunized received a prescription for Tdap-IPV. Telephone interview to all the participants at 8–10 weeks after discharge.	Among the women unimmunized at delivery, the percentage vaccinated postpartum climbed from 17 to 42% between 2011 and 2015, while the percentage of their unimmunized partners who were vaccinated remained stable (27 and 29%).
Healy et al. [44]	USA, 2011	Tdap	- Phase 1 (January 2008–January 2010): 11.174 postpartum women, largely underinsured, medically underserved, population. - Phase 2 (January 2009–January 2010): the program was expanded to1860 family contacts.	Observational study. A questionnaire was provided asking for: - personal demographic data - previous Tdap vaccination - medical contraindications for Tdap	- Education for healthcare professionals and postpartum women and families: posters and program education incorporated into antenatal, baby-care, and breastfeeding classes and written information packet; - Nurses available to answer questions; - In Phases 1 and 2, a standing order applied for Tdap vaccination of postpartum women; - Women who consented also received a rubella vaccine at a different site, if the latter was indicated; - In Phase 2, the vaccination was expanded to family contacts who consented.	- 8334 (75%) of 11,174 postpartum women received Tdap; - A median of 2 (range, 0–10) contacts per infant received Tdap vaccine; - 1697 (91%) received Tdap vaccine before infant hospital discharge, and 144 (8%) received Tdap vaccine within 7 days after hospital discharge.

^1^ Tetanus toxoid, reduced diphtheria toxoid and acellular pertussis vaccine. ^2^ Tetanus toxoid, reduced diphtheria toxoid, acellular pertussis and poliomyelitis vaccine. ^3^ New South Wales.

**Table 2 vaccines-12-01130-t002:** Studies presenting postpartum interventions to increase maternal vaccination and vaccinations of the other household members of the newborns with influenza with or without Tdap and COVID-19.

Reference	Country/ Year	Vaccine	Number of Participants and Characteristics	Study Design: Survey/Questionnaire	Intervention	Outcome
Hebballi et al. [45]	USA, 2022	Tdap, Influenza	200 postpartum women (June–August 2018)	Cross-sectional observational study. The survey included questions on: - Demographics, prenatal care; - Knowledge on Tdap, influenza vaccine; - Previous information on both vaccines; - Reasons for vaccine decline.	- Oral and written education of participants about Tdap vaccination, no counseling about influenza (non-influenza season); - Offer of Tdap to eligible participants; - A bedside nurse was notified if the patient was willing to be vaccinated prior to discharge; - No offer of influenza vaccine (non-influenza season).	- 97 eligible participants, 25% were vaccinated before the survey, 38.2% after; - Doctor’s recommendation, infant protection, self-protection reported from those vaccinated; - The common barriers for non-immunized included a lack of vaccine offer by the provider and belief that vaccination was unnecessary.
Jordan et al. [31]	USA, 2015	Influenza	89,792 pregnant and postpartum women were approached, 28,609 responded to the first contact, 6841 completed the study. (October to November 2012)	Randomized control trial. Text4baby was used (free US national mobile health service) - Enrollees received a baseline survey via text asking whether they were planning to receive a flu vaccine this year (October 2012)	- Participants responding “yes” to the baseline survey (planners) randomly assigned to two groups and received either a “usual message” or an “enhanced message’’: one message plus the opportunity to set up a reminder. - Participants who responded “no” to the baseline survey (non- planners) were randomly assigned to two groups and received either the “usual message” or the “enhanced message” (were asked why not planning to receive the vaccine plus educational message for the vaccine).	“Planners” - Among both pregnant and postpartum women, receipt of an enhanced reminder increased the odds of influenza vaccination at follow-up or intention to do so. “Non-planners” - The receipt of the enhanced message was not associated with receipt of the vaccine or intention to do so.
Maltezou et al. [46]	Greece, 2012	Influenza	- 224 postpartum women who delivered in a maternity hospital or whose neonate was admitted to a neonatal unit in Athens. - 224 fathers of the neonates. (November 2011 to February 2012)	Observational study.Demographics, epidemiologic, clinical, pregnancy, and birth data were collected using one standardized form per mother.	- Mothers, fathers and household members were informed about recommendations for influenza vaccination of family members and household contacts of infants, efficacy and safety of influenza vaccine, and expected effectiveness to their baby; - All mothers were offered the vaccine (on the ward, free of charge) as well as fathers and household members; - Those who refused vaccination were asked to provide the reason for doing so.	Of the 224 mothers, 165 (73.7%) received influenza vaccine prior to discharge from the hospital. Of the 224 fathers, 125 received the influenza vaccine (55.8% vaccination rate); 51 (22.7%) of 224 families had all household contacts vaccinated against influenza (complete cocoon).
Kouba et al. [36]	USA, 2022	COVID-19	8281 unvaccinated postpartum women during delivery hospitalization at seven hospitals in New York (May 2021–September 2021)	Retrospective cohort study. Sociodemographic characteristics were obtained from medical records	- Offer of COVID-19 vaccine before hospital discharge.	412 of the 8281 unvaccinated postpartum women received the vaccine (5%).
Momani et al. [35]	Jordan, 2023	COVID-19	425 women unvaccinated for COVID-19 vaccine (December 2021–April 2022): They were breastfeeding women, pregnant or planning to be pregnant separated into: - 205 intervention group (33 of them breast-feeding); - 220 women control group (46 of them breast-feeding).	Prospective controlled trial. - Demographics, previous COVID-19 infection, educational and financial status; - Questionnaire assessing COVID-19 hesitancy.	- The women in the intervention group received individual tele-education: interactive phone consulting sessions, text message, digital education booklet. - The women in the control group did not receive education.	- Intervention group: 162/205 women received the vaccine (79%). - Control group: 4/220 women were vaccinated (2%). Not clarified how many of these women were recruited in the postnatal period.

## Data Availability

Data available on request.

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
