# Peer review of "Postpartum Interventions to Increase Maternal Vaccination Uptake: Is It Worth It?"

_vaccines, 2024, doi:10.3390/vaccines12101130_

Round 1

Reviewer 1 Report

Comments and Suggestions for Authors

Dear colleagues,

Thank you very much for submitting your manuscript. 

The study of postpartum interventions to increase maternal vaccination uptake is an important subject that could provide information aimed at improving the approach and care of these categories of patients.

The authors presented the search results related to the vaccine against Pertussis, influenza, and COVID-19. The search interval was chosen from January 2009 to January 2024. The anti-COVID-19 vaccine was only available on the market in 2020. I think it is not a good idea to do a study on a time interval in which there was no vaccine. In addition, the search results in this case highlighted only two studies, which cannot lead to relevant conclusions.

While the literature search was diligently performed in Medline, I believe a more comprehensive approach involving multiple databases would greatly enrich the study's findings and enhance its credibility. 

The presented aspects need to support the conclusions sufficiently.

Author Response

Dear colleagues,

Thank you very much for submitting your manuscript. 

The study of postpartum interventions to increase maternal vaccination uptake is an important subject that could provide information aimed at improving the approach and care of these categories of patients.

We would like to take this opportunity to thank the reviewer for the positive feedback and comments to improve our manuscript further.

The authors presented the search results related to the vaccine against Pertussis, influenza, and COVID-19. The search interval was chosen from January 2009 to January 2024. The anti-COVID-19 vaccine was only available on the market in 2020. I think it is not a good idea to do a study on a time interval in which there was no vaccine. In addition, the search results in this case highlighted only two studies, which cannot lead to relevant conclusions.

This is a very interesting point and worth mentioning. As COVID-19 vaccine was only available in 2020 there were very few studies in the field and we need to acknowledge this as one of the limitations of the present work. As rightly highlighted by the reviewer we cannot draw conclusions based on only few studies yet we may see a trend similar to the one observed for other vaccinations (pertussis, influenza) recommended for women during the postnatal period.

While the literature search was diligently performed in Medline, I believe a more comprehensive approach involving multiple databases would greatly enrich the study's findings and enhance its credibility. 

This is a point worth acknowledging in the limitations session of our manuscript as we might have missed studies published in other databases. Yet, we believe that high quality studies are usually published in the MEDLINE and therefore the conclusions from the present systematic review are credible. However, if the Editor and Reviewer wish as to do so we would be delighted to perform an extra search

Following your above suggestions we have added the following sentences to the limitation session:

Moreover, our search was performed in the MEDLINE hence we might have missed studies published in other databases. Finally, the COVID19 vaccine was licensed for use in 2020 and therefore very few data are available in the field compared to the other two vaccines recommended for years during pregnancy and postnatal period.

Reviewer 2 Report

Comments and Suggestions for Authors

A well researched article with clear descriptions of each article reviewed. 

I would have liked to see some recommendations in the discussion section based on the reviewed articles - perhaps summarising the options and making an informed decision which way health care professionals can go

Author Response

A well researched article with clear descriptions of each article reviewed. 

We would like to take this opportunity to thank the reviewer for the positive feedback and comments to improve our manuscript further.

I would have liked to see some recommendations in the discussion section based on the reviewed articles - perhaps summarising the options and making an informed decision which way health care professionals can go.

Thank you very much for the useful comments. We have revised the discussion section based on the results of the reviewed articles and according to these results we concluded to suggestions for healthcare professionals use in order to promote successfully maternal vaccination.

Reviewer 3 Report

Comments and Suggestions for Authors

General comment

The paper “Postpartum interventions to increase maternal vaccination up-2 take. Is it worth it?” is an interesting article about the evidence of interventions during the postpartum period to improve maternal vaccination and indirectly protect newborns and young infants but the article needs significant editorial revision.

Major comments

The authors provide the protocol record in Prospero but the methods are not explained in sufficient detail. Lack definitions of outcome and main independent variable and the authors should explain in more detail the inclusion and exclusion criteria, study design, the variables included in each article, the methods for summarizing the information, and the quality control assessment of each study.

Specific comments

1)      Abstract. Authors should be added delate superficial information and include objective, basic methods, main results and conclusion basic in the main result.

2)      Introduction: explain background on different vaccination coverage by each vaccine (both in pregnancy and postpartum)

3)      Methods: adds standard information used to conduct a systematic revision (definitions of outcome, independent variable,  inclusion and exclusion criteria, the variables included in each article, vaccine hesitancy, study design, the methods for summarizing the information, and the quality control assessment of each study.

4)      Results: explain de different study design (pre-post without random assignment, intervention with or without group control, …); improve description of results to the main results (the role of healthcare providers, administration of vaccine in the health services …), complete information in  table 1

5)      Discussion: comment the different results by vaccination, study deign, definition of outcome. The final reference to respiratory syncytial virus mis not supported by this study. Review the whole conclusion and recommendations.

6)      Check the reference style. There are important errors (ie: ref 27)

Author Response

Reviewer 3

General comment

The paper “Postpartum interventions to increase maternal vaccination up-2 take. Is it worth it?” is an interesting article about the evidence of interventions during the postpartum period to improve maternal vaccination and indirectly protect newborns and young infants but the article needs significant editorial revision.

We would like to take this opportunity to thank the reviewer for the positive feedback and suggestions to improve our manuscript further

Major comments

The authors provide the protocol record in Prospero but the methods are not explained in sufficient detail. Lack definitions of outcome and main independent variable and the authors should explain in more detail the inclusion and exclusion criteria, study design, the variables included in each article, the methods for summarizing the information, and the quality control assessment of each study.

Thank you for the comment.

Following your above suggestions we have added the following paragraphs to the methods session to clarify our methodology:

Studies including data about pertussis/influenza/COVID-19 but irrelevant to vaccination were excluded from this review. Studies including data about vaccination uptake in the postpartum period but not including some type of intervention were excluded. Intervention trials not aiming to increase vaccination uptake in the postpartum period were also excluded.

The studies included aimed to promote vaccination with pertussis, influenza, COVID-19 or combination of these vaccines. The population of the studies were postpartum women and other household contacts of the newborns. The interventions applied to this population in the maternity ward before discharge or in their first postpartum visits or in their first visits to the pediatric office soon after discharge from the maternity ward. Studies included only pregnant women or other population were excluded.

The intervention applied in the studies included information and education of the postpartum women, offer of the vaccine to the postpartum women and other household contacts of the newborns, education of the health care providers caring for postpartum women and newborns and any other measures applied promoting postpartum vaccination.

The outcome of the studies included were differences observed in vaccination uptake rates between the intervention and control arm or between the pre and post-intervention period

For the quality control assessment of each individual study please refer to e-table (supplemental table) available after the references.

Specific comments

  1. Authors should delete superficial information and include objective, basic methods, main results and conclusion basic in the main result.

Thank you for the comment. We have now amended the whole abstract paragraph following your advice.

Vaccination of pregnant and postpartum women for pertussis, influenza and COVID-19 protects not only themselves but their offspring as well. Despite the benefits of this approach, vaccination uptake remains suboptimal in pregnancy. Where the opportunity to be vaccinated in pregnancy is missed, the offer of vaccination in the post-partum period may be an alternative strategy. The aim of this systematic review is to assess the impact of interventions to increase vaccination uptake in the postpartum period on vaccination uptake. Therefore, a literature search was performed in MEDLINE, including interventional studies promoting vaccination uptake in postpartum women published between 2009 and 2024. The search was conducted according to PRISMA guidelines and registered with PROSPERO. We finally included 16 studies in the review and the primary outcome was vaccination uptake in the postpartum period. The most important factors for increasing uptake were recommendation from healthcare providers, type of interventions used, and delivery of vaccines in the maternity wards or the community. In conclusion, maternal vaccination rates in the postpartum period may increase with targeted education by healthcare professionals and positive reinforcement. The interventions described in these studies could be applied in the healthcare systems worldwide.

  1. Introduction: explain background on different vaccination coverage by each vaccine (both in pregnancy and postpartum)

Thank you for the very useful comment. As you suggested we have given some extra data on maternal vaccination primarily during pregnancy as no data on postpartum vaccination coverage is available. We added this extra paragraph in the introduction in order to explain the vaccine coverage for each vaccine:

Despite the importance of vaccination for pregnant and postpartum women and the current guidelines, vaccination coverage among pregnant women has been suboptimal. Data from the Centers for Disease Control and Prevention (CDC) showed a 45,8 % vaccination rate for Tdap and 48,4% for influenza during pregnancy. Receipt of both flu and Tdap vaccines was reported by 22.6% of women with a recent live birth, a decrease from the previous season (30.7%).  [25] As for COVID-19, the vaccination rates in pregnancy ranges from 27 to 31%. [26, 27]

  1. Methods: adds standard information used to conduct a systematic revision (definitions of outcome, independent variable,  inclusion and exclusion criteria, the variables included in each article, vaccine hesitancy, study design, the methods for summarizing the information, and the quality control assessment of each study.

Thank you for the comment.

Please refer to the response to first comment as well as the revised tables 1,2 with more information included on type of studies included  as well as the e-table(supplementary) table for the quality assessment control.

For clarification all the included studies were interventional studies and the additional categorization of the studies was added in Tables 1 and 2.

According to your advice we added this extra paragraph in the beginning of the result section:

In Tables 1 and 2 we present the variables included in each article such as the number of participants and the characteristics of them, whether a questionnaire was prior to the intervention process, as well as the type of intervention applied to each one of these studies. Most importantly, the effect of these interventions to the vaccination uptake in postpartum period is also presented. A summary of the 16 studies is provided below. The studies are mainly discussed and categorized according to the primary outcomes and the type of interventions applied. Finally some supplemental outcomes of the studies are discussed.

  1. Results: explain the different study design (pre-post without random assignment, intervention with or without group control, …); improve description of results to the main results (the role of healthcare providers, administration of vaccine in the health services …), complete information in  table 1

Please refer to the revised tables 1 and 2 where more information on study type and interventions is provided

We have also changed the subheading of the results 3.2, 3.2.1, 3.2.2, .3.2.3, 3.2.4, 3.2.5, 3.2.6.for further clarification as follows:

3.2. Summary of studies according the results and type of intervention

3.2.1. Primary outcome according to study design

3.2.2. Type of intervention – The role of different healthcare professionals providing information

 3.2.3. Type of intervention – The role of different methods of educational process

 3.2.4. Type of intervention – The role of health care provider’s education as a method of intervention

 3.2.5. Type of intervention – The role of vaccine offer

3.2.6. Supplemental outcomes – Reasons for vaccine hesitancy.

  1. Discussion: comment the different results by vaccination, study deign, definition of outcome. The final reference to respiratory syncytial virus mis not supported by this study. Review the whole conclusion and recommendations.

Thank you for your insightful comments. We reviewed the whole conclusion according your recommendations. We added an extra paragraph (2nd paragraph) and added some information in the third paragraph in order to comment the different results by vaccination, study design and definition of outcome.

We deleted the final reference to respiratory syncytial virus.

  1. Check the reference style. There are important errors (ie: ref 27)

Thank you for your correction. We revised and checked the reference style.